# Implant-Related Complications Do Not Interfere with Corrections with the Shilla Technique in Early Onset Scoliosis: Preliminary Results

**DOI:** 10.3390/children10060947

**Published:** 2023-05-26

**Authors:** Mehmet Bülent Balioğlu, Kadir Abul, Ahmet Onur Akpolat, Ali Volkan Özlük, Nurullah Saçık, Mehmet Fatih Aksay, Mehmet Çetinkaya

**Affiliations:** 1Department of Orthopedics and Traumatology, Başakşehir Pine and Sakura City Hospital, 34480 Istanbul, Turkey; 2Department of Orthopedics and Traumatology, Fatih Sultan Mehmet Training and Research Hospital, 34752 Istanbul, Turkey

**Keywords:** growing spine, Shilla technique, growth guidance, apical fusion, active apex correction

## Abstract

Growth-preservation techniques are utilized in early onset scoliosis (EOS) cases requiring surgical intervention. The Shilla technique corrects the deformity by reducing additional surgeries with its growth-guidance effect. As with other techniques, various problems can be encountered following the administration of the Shilla technique. The aim of this study was to examine the effect of complications encountered with the Shilla treatment on correction and growth. Sixteen patients with a follow-up period of at least one year after receiving Shilla growth guidance for EOS were included in this retrospective study. No complications occurred, and no unplanned surgery was required in 50% of the cases. Of the remaining eight patients with postoperative implant-related complications (50%), six (37.5%) required unplanned surgery; this consequently caused implant failure in the proximal region in five cases (31.25%) and deep tissue infection around the implant in one case (6.25%). Deformity correction, spine length, and quality-of-life scores significantly improved in EOS through Shilla growth guidance. In terms of spinal growth and deformity correction, there were no significant differences between patients with implant-related problems and individuals without occurrences. Although implant-related problems were detected in our dataset and corresponding unexpected surgeries were necessary, these complications had no significant unfavorable influence on correction and spine growth.

## 1. Introduction

Early onset scoliosis (EOS) is a progressive condition characterized by complex curvatures of the pediatric spine caused by a variety of etiologies (idiopathic, congenital, syndromic, and neuromuscular) in children under the age of ten. If left untreated, these patients may suffer poor respiratory and cardiac development, progressive deformities, and complex and difficult-to-treat spinal problems [1,2,3]. Currently, surgical treatments with growth-preserving instrumentation (GPI) techniques without growth arrest are preferred in cases of high-angle and progressive EOS. Several surgical methods have been reported in the literature; these include traditional growing rods (TGR), vertically expandable titanium ribs (VEPTR), and magnetically controlled growing rods (MCGR) applied for their spinal ‘distraction’ effects. The Shilla technique (Shilla) and the Luque Trolley technique (Modern Luque Trolley) are applied owing to their ‘growth-guidance’ effects and applications in vertebral body tethering (VBT), which are fairly new modalities applied to the convex side of the curvature and whose compressive ‘growth modulation’ effect is utilized [4,5,6,7,8,9]. The Shilla procedure has an advantage over other GPI techniques since it uses passive growth guidance to correct the deformity and does not necessitate planned surgeries in most cases [6,8,9,10]. Patients can be observed for an extended period of time without surgery due to the fixation of the apical vertebral segments and the non-locking of the screw heads positioned proximal and distal to the instrumentation [11,12].

Problems related to wound healing, infection, implant failure, metallosis, early fusion, neurologic deterioration, and unplanned surgeries may occur in all EOS cases treated using GPI, regardless of the technique used. When determining the best GPI for the patient, the goal of surgery should be to avoid all-time complications. Many multicenter studies have compared the advantages and disadvantages of different growth-friendly techniques [1,13,14,15]. Even though various problems may also occur as a result of Shilla application, this technique aims to avoid these problems and those caused by active lengthening procedures.

Research on the impact of implant-related complications associated with Shilla growth guidance on the correction of patients’ deformities and growth remains limited, and additional investigations are necessary to evaluate the influence of these complications on treatment outcomes. Therefore, we conducted a comprehensive analysis of the complications that arise from using the Shilla growth guidance technique in patients with EOS of diverse etiologies, along with the adverse effects that result from these complications. Moreover, we investigated whether implant-related complications may hinder the correction of deformities and impede growth in patients treated with the Shilla technique.

## 2. Materials and Methods

This clinical study involved a total of 26 patients who underwent Shilla treatment for EOS with varying etiologies between January 2013 and December 2021, and a retrospective analysis was carried out. The investigation was conducted at three distinct academic hospitals under the direction of the responsible surgeon. Among these patients, 16 were selected for inclusion in this case–control series, as they were followed up for a minimum of one year (30.8 ± 29.2 months) and met the inclusion criteria. The study analyzed the improvement in deformity and growth effects, along with implant-related complications and unplanned surgeries. Interpatient comparisons were performed between those who experienced implant-related complications during treatment and those who did not. While three patients (18.8%) underwent final graduation fusion surgery, treatment continued for 13 patients. Cobb angles, spine length between T1–T12 and T1–S1, kyphosis (T4–12) and lordosis (L1–S1) angles in the sagittal plane, coronal apical vertebral levels, coronal, sagittal, pelvic, and shoulder balances, and EOSQ-24 (Early Onset Scoliosis 24-Item Questionnaire) results were collected preoperatively, early postoperatively, and at the final follow-up. Inclusion criteria were as follows: (1) age between 5 and 10 years, (2) Cobb curve angle of >45°, (3) increase in Cobb angle of >10° within 1 year, (4) open triradiate cartilage, (5) Sanders stage of less than 3, (6) unsuccessful conservative treatment, (7) no previous deformity surgery, and (8) regular follow-up >1 year by the responsible author. Exclusion criteria were as follows (1) age of <5 years and >10 years, (2) Cobb angle of <45°, (3) closed triradiate cartilage, (4) Sanders stage of greater than or equal to 3, (5) improvement in curvature with conservative treatment, (6) previous spinal surgery, and (7) no regular follow-up.

Demographic characteristics, etiology, follow-up periods, complications, and magnetic resonance imaging (MRI) results were recorded. For radiological evaluation, standard whole spinal orthoroentgenography was obtained preoperatively, in the early postoperative period, and at the last follow-up examination in the standing position. Additional seated position X-rays for neuromuscular and syndromic patients who were unable to stand were also obtained. The entire spinal region was evaluated by a radiologist experienced in spinal radiology with MRI for preoperative intraspinal pathology control (Table 1). Ethics committee approval was obtained from our hospital (KAEK/2021.12.296).

### Statistical Analysis

The mean, standard deviation, median, minimum, maximum, frequency, and ratio values were used to present the descriptive statistics. A Kolmogorov–Smirnov test was used to check the conformity of the variables to a normal distribution. Independent sample *t*-tests and Mann–Whitney U tests were used to analyze the independent quantitative data. Paired sample *t*-tests and Wilcoxon tests were used to analyze the dependent quantitative data. Spearman correlation analysis was used for correlation analysis. The SPSS 28.0 program was used for the analysis. The results were measured and compared by two spinal surgeons.

## 3. Results

In measurements recorded during the early postoperative period and final follow-up, improvements in the Cobb angle and shoulder balance angle, and increases in T1–T12 and T1–S1 length were significant compared with preoperative measurements (*p* = 0.005, *p* = 0.002, *p*= 0.002, *p* = 0.055). No significant change was observed in terms of kyphosis (T4–T12), lordosis (L1–S1), coronal balance, sagittal balance, and pelvic balance (*p* = 0.258, *p* = 0.659, *p*= 0.182, *p* = 0.831, *p* = 0.139) (Table 2). Intra- and inter-correlations were high (intraclass correlation coefficient >0.80, *p* < 0.05).

No significant intergroup difference was observed in measurements recorded in the preoperative period, early postoperative period, and final follow-up in terms of the Cobb angles and T1–T12 and T1–S1 length (*p* = 0.832, *p* = 0.916, *p* = 0.949) (Table 3 and Table 4).

In the early postoperative period and final follow-up measurements, the group with complications showed significantly lower T1–S1 values than the group without complications (*p* = 0.038, *p* = 0.040) (Table 5).

Kyphosis (T4–T12) values during the preoperative period, early postoperative period, and final follow-up showed no significant intergroup differences (*p* = 0.704). The group with complications showed significantly lower lordosis (L1–S1) values in the postoperative and final follow-up measurements than the group without complications (*p* = 0.659). The decrease in lordosis between preoperative and early postoperative measurements was significantly higher in the group with complications than in the group without complications (*p* = 0.030). In the early postoperative period and final follow-up measurements, no significant intergroup difference was observed in terms of the lordosis values (*p* = 0.422). In the preoperative, early postoperative period, and final follow-up measurements of coronal, sagittal, shoulder, and pelvic balance, no significant intergroup differences were noted (*p* = 0.149, *p* = 0.418, *p* = 0.606, *p* = 635). In the group with complications, the final follow-up measurement of shoulder balance showed a significant decrease compared with the preoperative period (*p* = 0.037). The change in preoperative and early postoperative period shoulder balance measurements showed no significant differences between the groups with and without complications (*p* = 0.684). The change in early postoperative period and final follow-up shoulder balance measurements was not significantly different between the groups with and without complications (*p* = 0.827). In the group with complications, early pelvic balance showed a significant decrease in terms of the values measured in the postoperative and preoperative periods (*p* = 0.030). All components of the EOSQ-24 score (*p* < 0.05), except the daily living domain (*p* = 0.539), showed improvements in the postoperative period (Table 6).

In eight patients (50%) who underwent Shilla treatment for EOS, no complications developed during the treatment period, spinal growth and curve correction were not adversely affected, and no unplanned surgery was required. Postoperative implant-related complications such as proximal screw loosening, screw malposition, proximal junctional kyphosis (PJK), wound problems, rod breakage, metallosis, and infection were observed in eight patients (50%). Six (37.5%) of the eight patients with implant-related problems required unplanned surgery. In five cases (31.25%), unplanned surgery was performed to repair implant failure in the proximal region, and in a single case (6.25%), surgery was performed due to deep tissue infection around the implant. Unplanned surgeries were performed four times in one patient, twice in another, and once in four others. The etiologies and index surgical age distributions of the six individuals who underwent unplanned surgery showed that two were idiopathic (ages 115 and 116 months), three were syndromic (Marfan; 60, and 55 months, Ch. 6 deletion; 55 months, Doose; 72 months), and one had NMS (cerebral palsy (CP); 61 months). Loosening of the proximal pedicle screws, PJK, and skin irritation were observed in five cases, whereas four cases developed wounds as a result of skin irritation. Metallosis induced by movement in the region of the implant in the screw–rod connection area was seen in all patients needing unplanned surgery. The highest number of unplanned surgeries was four in a patient with chromosome 6 deletion syndrome (55-month-old male, follow-up: 26 months). The accompanying kyphosis deformity (thoracic proximal kyphosis of 62°) and inadequate bone morphology were the causes of early loosening and failure of the proximal screw fixation. Although pedicle screw fixation and rod placement provided initial improvement, early proximal implant failure, rod displacement, skin wound problems, and PJK resulted in repeated unplanned surgical interventions in that case. In the long term, maintenance of growth was planned while concurrently ensuring correction of the proximal kyphosis deformity at the appropriate stage. In a patient with Marfan syndrome, high-angle scoliosis, and kyphosis deformity (60 months, female, follow-up 12 months), dislocation of the proximal screw required unplanned surgery as a result of implant failure (Figure 1).

In a patient with NMS-CP (61-month-old male, follow-up: 16 months), unplanned surgery was performed 3 months after the index surgery due to screw–rod loosening and rod displacement caused by dislocation of screw caps in the concave apical region with proximal failure. Infection emerged in one case (6.25%) (72-month-old girl with Doose syndrome, follow-up period: over 12 months) after a respiratory tract infection in the peri-implant deep tissue five months after surgery due to low body resistance. The deep-wound infection resolved after peri-implant debridement and antibiotherapy according to the recommendations of the infectious disease unit, and no additional problems were encountered in the follow-up. Rod breakage was observed in two cases (12.5%) unilaterally on the concave side of the curve immediately distal to the apical fusion site. No additional unplanned surgeries for unilateral rod breakage were required in either case. An 8-year-old girl with NF-1 continued to be followed up uneventfully in the 4th year of treatment (Figure 2). A patient with Down syndrome did not require any intervention during 8 years of follow-up and did not require additional surgery despite unilateral rod breakage. Planned graduation surgery with instrumented fusion was uneventful, but the patient died due to sudden cardiac arrest just after postoperative CVP catheter removal on the day of discharge from the hospital.

The treatment of the patients who underwent surgery for EOS was accompanied by several additional challenges in terms of anesthesia and reanimation. Video laryngoscope guidance was needed in two of our patients (12.5%) because of difficult intubation. Postoperative respiratory problems were observed in three cases (18.75%). Prolonged treatment in the postoperative intensive care unit and the ward was required. The early difficulties encountered in the administration of anesthesia and normalization of postoperative respiration were overcome with good care and rehabilitation. In all patients, neuromonitoring was used in index surgeries and when additional surgery was required; no neurological changes were observed during these procedures.

## 4. Discussion

In the present study, whether implant-related problems had a negative effect on deformity correction and spinal growth was investigated, and unplanned surgeries and complications were retrospectively evaluated. The incidence and prevalence of EOS are estimated to be 0.27 and 1.07 per 10,000 children, respectively, in a recent regional multicenter epidemiological study by AlNouri et al. [16]. EOS affects males more than girls, and the majority of cases are idiopathic or related with underlying neuromuscular or syndromic disorders. EOS can have serious physical, psychological, and social repercussions that impact patients’ quality of life and ability to function. Notarnicola et al. [17] found that teenagers with scoliosis had lower levels of physical activity and perceived a bigger impact of their disease on their everyday lives than their non-scoliotic counterparts in a recent observational study. Early detection and care are critical for preventing the advancement of EOS and improving patient outcomes.

Increasingly intensive efforts have been undertaken in EOS treatment in the last 15 years owing to improvements in treatment algorithms and GPI techniques [6,7,8,10,18,19,20]. One of these methods, the Shilla technique, is recommended as a method that guides growth by reducing the number of surgical interventions while facilitating the correction of the deformity [11,12]. The results and complication rates of EOS treatment have been reported in various studies [15,21,22,23,24]. Reportedly, the Shilla procedure is a suitable option to treat EOS with different etiologies; however, this procedure is also associated with moderate complication rates [12,22]. Wilkinson et al. [22] reported an overall implant-related complication rate of 29% after the Shilla procedure with a minimum follow-up of 5 years. Screw pull-out and rod breakage occurred in 14% and 14% of the patients, respectively, and deep and superficial infections developed in 9.5% and 4.7% of the patients. In the literature, the Shilla growth guidance technique is an alternative to distraction-based growing rod (GR) systems [12]. Comparing both applications, Andras et al. showed a greater improvement in the Cobb angle and a greater increase in T1–S1 length in patients who underwent GR compared to Shilla [14]. In a study comparing the radiologic results of TGR and Shilla treatment, Luhmann et al. did not report a significant intergroup difference in clinical parameters during follow-up, finding that the number of operations was significantly higher in the TGR group [15]. Loss of correction and the need for osteotomy were reported as disadvantages of Shilla treatment and concave side osteotomies may be required, leading to serious complications [25]. Loss of correction via crankshafting or adding-on (especially distal migration of the major curve) was reported in a significant percentage of patients undergoing Shilla growth guidance [22,26]. Agarwal et al. described the modified Shilla approach (active apex correction: APC) as a solution that can help dynamically re-modulate the apex of the deformity and reduce correction loss. A modified Shilla approach was compared with the TGR system, and no significant intergroup difference was noted in terms of correction and growth. Biomechanical complications were more frequent in the GR system (13 of 26 cases) compared to APC (5 of 20 cases). Surgeries related to active extensions with TGR and a longer follow-up period were thought to be effective [25]. In the literature, several comparative studies with other growth-friendly techniques have been conducted to better evaluate the efficacy of treatment in EOS. Haapala et al. [1] compared MCGR with the Shilla technique and reported similar EOSQ-24 results between the two methods. Furthermore, the lowest postoperative scores were observed in the daily living domain in both treatment groups. In our patients, we observed significant improvement in all domains of EOSQ-24 results after Shilla treatment compared to pre-treatment scores, except for the daily living domain as the prementioned studies.

Children with EOS are prone to develop pulmonary complications after repeated anesthesia procedures. Difficult intubation may be observed more frequently, especially in neuromuscular and syndromic cases [24]. Furthermore, the potentially harmful effects of frequent anesthesia exposure are not yet fully understood.

EOS surgeries typically result in proximal junctional kyphosis and proximal anchor problems. Avoiding extensive thoracic kyphosis correction during growing rod treatments is indicated in order to reduce the risk of PJK [27]. We believe that in our one example of chromosome 6 deletion syndrome, the aggressive correction of the proximal kyphosis driven by the T6 anterior wedge vertebra was directly related to the failure of the proximal anchors.

The weakening of the posterior ligamentous tissues may also be related to early implant failure and adverse outcomes such as implant loosening and displacement seen in the upper vertebral levels where the implant is inserted. By generating kyphosis, intraoperative posterior distraction forces also can contribute to the prevalence of PJK. El-Hawary et al. found that immediately following implantation and at a minimum 2-year follow-up, the chance of developing PJK with distraction-based growth-friendly treatment for EOS was 20% and 28%, respectively. Both the rib-based therapy and the spine-based therapy groups shared the same chance of developing PJK. In comparison to the non-PJK group, their cases with PJK preoperative were considerably more thoracic kyphosed and were often older (7.1 y PJK vs. 5.0 y no PJK). Preoperative pelvic tilt and lumbar lordosis in the PJK group tended to be greater than in the non-PJK group. It was suggested that the patient’s family be informed of the potential risk of developing PJK as a result of having EOS therapy during preoperative patient education [28].

There is limited information on the success rates of different treatment options for proximal junctional problems in EOS. Several factors can affect the success rates of treatment for proximal junctional problems in EOS. The location of the upper instrumented vertebra in relation to the sagittal apex is an important factor that can affect the risk of PJK [29]. We also experienced a high apex kyphosis causing upper anchor problems in one particular case (Deletion syndrome). Additionally, the technique used for instrumentation can also affect the success rates of treatment [30]. Miladi et al. found fewer mechanical complications with fusionless double ends technique using hooks instead of transpedicular screws for proximal fixation in a growth-friendly scoliosis surgery [29]. Other factors that can affect the success rates of treatment include the severity of the condition, the age of the patient, and the presence of any underlying conditions especially in neuromuscular origin [28,31] Further research is needed to determine the success rates of different treatment options for proximal junctional problems in EOS.

In the literature of implant-related complications, screw dislocation reportedly occurs in 31.25% [5], skin wound problems occurs in 25% [4], rod breakage occurs in 12.5% [2], and deep infection occurs in 6.25% [1] of patients. In our study, a total of 10 unplanned surgeries were performed: one in four patients, two in one patient, and four in one patient. In patients requiring surgery, signs of metallosis were observed in the rod–screw junction. Additionally, in our series, we observed difficult intubation in two (12.5%) syndromic cases and prolonged postoperative intensive care unit stay in three (18.7%) syndromic cases.

In the present study, pre- and postoperative data were evaluated retrospectively. Patients were operated and followed-up by the author from the indication to the final follow-up. Complications encountered during the follow-up period were closely monitored and managed in the same manner. The long treatment process and patients dropping out of follow-up for various reasons resulted in a decrease in the number of patients and a shortened follow-up period. The lack of comparative studies with other growth-friendly surgical techniques and short follow-up constituted the main limitations of our study. Scoliosis deformity was significantly improved compared to preoperative measurements, spine length was significantly increased, and quality-of-life scores also improved significantly. The analyses performed to evaluate the effect of complications showed that the lordosis angle and T1–S1 length were significantly lower in the early and final postoperative controls of those who developed complications compared to those who did not. Although Shilla treatment provided significant improvement in deformity correction and spine height compared to the preoperative period, implant-related complications had a relatively negative effect on spine height and lumbar lordosis.

## 5. Conclusions

Various implant-related problems were encountered in patients with EOS treated with the Shilla technique. Although these complications were observed after the Shilla procedure, an improvement in the correction of the initial deformity and a significant increase in spinal height were possible. When patients with implant-related complications were compared to patients without events, there were no significant differences between the two groups in terms of spinal growth and deformity correction. The T1–S1 length and lumbar lordosis appeared to be affected, especially in patients with complications compared to uncomplicated patients. In conclusion, it should be kept in mind that careful planning and long-term follow-up will be required when deciding on treatment with the Shilla method in EOS patients. It is crucial for successful treatment to avoid complications and minimize their negative effects. Therefore, close follow-up of patients and management of treatment by an experienced team are recommended.

## Figures and Tables

**Figure 1 children-10-00947-f001:**
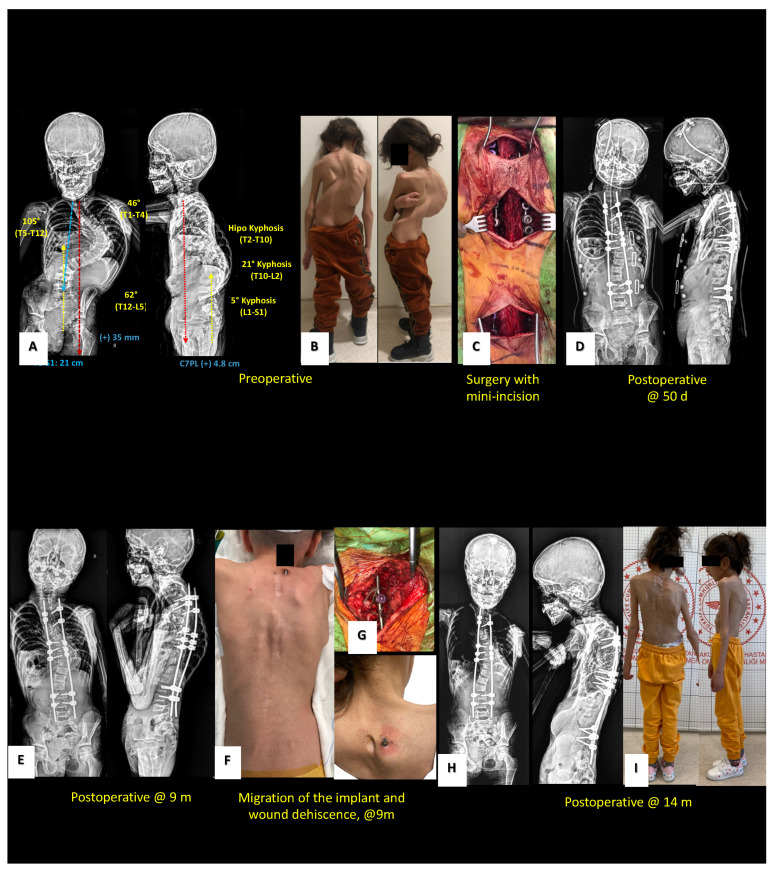
Preoperative radiological (**A**) and clinical (**B**) views of a 6-year-old girl with syndromic (Marfan) kyphoscoliosis. The case was operated with the Shilla technique using a minimal surgical incision approach. (**C**) Early postoperative X-ray images are shown. (**D**) Pediatric prolonged intensive care unit treatment was required because of postoperative respiratory distress. At the ninth postoperative month, proximal screw loosening and skin dehiscence occurred as a result of irritation of the skin by the rod tip (**E**,**F**). The rod was supported with sublaminar wires in the proximal region, and revision was performed without interrupting the remaining spinal growth (**G**). Follow-up continued after the 14-month period (**H**,**I**).

**Figure 2 children-10-00947-f002:**
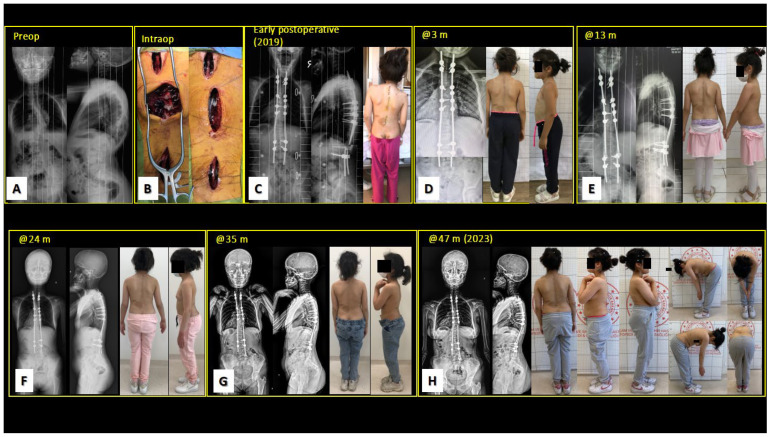
(**A**) Preoperative X-ray images of a 10-year-old girl with syndromic (NF-1) EOS. (**B**) Shilla growth guidance surgery was performed. (**C**–**H**) Follow-up period was 47 months and no additional surgery was required despite the development of concave rod breakage. Growth of the patient continued, and the case continued to be followed up.

**Table 1 children-10-00947-t001:** Demographic characteristics of the patients, etiology, follow-up periods, and complications encountered.

			Min–Max	Median	Mean ± SD/n-%
Follow-up Period (Month)			12.0–104.0	17.0	30.8 ± 29.2
Index surgery (Year)			4.5–12	9.4	8.8 ± 2.4
Gender	Female				11 (68.8%)
	Male				5 (31.2%)
Complication	(−)				8 ± 50.0
	(+)				8 ± 50.0
		Rod breakage (Exitus during CVP removal)			2 (1)
		Proximal screw loosening, PJK (skin wound dehiscence)			5 (4)
		Deep wound infection around the implant			1
Surgical Levels (n)			11.0–16.0	13.5	13.3 ± 1.36
Number of Moving Segments			7.0–11.0	9.0	8.94 ± 1.12
Etiology (n)	IS				7 (43.8%)
	Syndromic	NF	6		2 (12.5%)
		C6DS			1 (6.3%)
		DOS			1 (6.3%)
		DWS			1 (6.3%)
		MS			1 (6.3%)
	CS				1 (6.3%)
	NMS	NMS (SB)	2		1 (6.3%)
		NMS (CP, Syndromic *)			1 (6.3%)
MRI	Normal				14 (87.4%)
	NMS (SB, Arachnoid cyst)				1 (6.3%)
	CS ** (diastematomyelia, split cord, tethered cord, syringomyelia)				1 (6.3%)

(DWS: Down Syndrome, IS: Idiopathic Scoliosis, NF: Neurofibromatosis, C6DS: Chromosome 6 Deletion Syndrome, CS: Congenital Scoliosis, NMS: Neuromuscular Scoliosis, SB: Spina Bifida, MS: Marfan Syndrome, DOS: Doose Syndrome, PJK: Proximal Junctional Kyphosis, MRI: Magnetic Resonance Imaging). * Reciprocal translocation of the 8th and 20th chromosomes. ** Operated by neurosurgery for intraspinal pathology.

**Table 2 children-10-00947-t002:** Evaluation of preoperative, early postoperative, and final follow-up radiological measurements.

		Min–Max	Median	Mean ± SD	*p* *	*p* **
Cobb Angle (°)	Preop	47.0–103.0	61.0	64.6 ± 14.4		
	Early Postop	4.0–36.0	19.5	21.0 ± 9.9	0.000 ^E^	
	Final Follow-up	5.0–45.0	23.0	25.4 ± 12.6	0.000 ^E^	0.005 ^E^
T1-T12 (cm)	Preop	13.0–22.0	17.4	17.5 ± 2.7		
	Early Postop	15.3–24.0	19.4	20.0 ± 2.6	0.000 ^E^	
	Final Follow-up	17.4–25.0	22.0	21.3 ± 2.8	0.000 ^E^	0.002 ^E^
T1-S1 (cm)	Preop	20.6–35.6	28.0	28.6 ± 4.6		
	Early Postop	27.7–39.0	31.4	32.6 ± 3.9	0.000 ^E^	
	Final Follow-up	28.0–39.2	34.1	33.6 ± 3.8	0.000 ^E^	0.055 ^E^
Kyphosis (T4-T12) (°)	Preop	10.0–7	44.0	41.7 ± 19.8		
	Early Postop	15.0–62.0	36.5	36.4 ± 14.3	7 ^E^	
	Final Follow-up	15.0–60.0	35.5	37.3 ± 14.3	0.258 ^E^	7 ^E^
Lordosis (L1-S1) (°)	Preop	0.0–80.0	53.0	47.4 ± 22.0		
	Early Postop	7.0–62.0	43.0	40.4 ± 17.3	0.065 ^E^	
	Final Follow-up	15.0–65.0	42.0	41.7 ± 14.9	0.246 ^E^	0.659 ^E^
Coronal Balance (CSVL) (cm)	Preop	0.0–5.3	1.3	1.6 ± 1.8		
	Early Postop	0.0–5.9	1.1	1.5 ± 1.9	0.638 ^W^	
	Final Follow-up	0.0–5.0	1.5	1.8 ± 1.4	7 ^W^	0.182 ^W^
Sagittal Balance (cm)	Preop	0.0–8.0	3.4	4.0 ± 2.4		
	Early Postop	0.0–16.0	2.6	3.5 ± 4.3	0.600 ^E^	
	Final Follow-up	0.0–8.0	3.0	3.3 ± 2.2	0.230 ^E^	0.831 ^E^
Shoulder Balance (°)	Preop	1.0–19.0	5.5	6.3 ± 4.5		
	Early Postop	1.0–15.0	5.0	5.8 ± 3.8	7 ^E^	
	Final Follow-up	0.0–9.0	1.8	2.2 ± 2.4	0.003 ^E^	0.002 ^E^
Pelvic Balance (°)	Preop	0.0–15.0	3.0	4.0 ± 3.9		
	Early Postop	0.0–5.0	2.0	1.9 ± 1.5	0.028 ^W^	
	Final Follow-up	0.0–8.5	2.0	2.4 ± 2.3	0.139 ^W^	7 ^W^

^E^ Paired samples *t*-test/^w^ Wilcoxon test. *p* * Comparison to preoperative period/ *p* ** Comparison to early postoperative period.

**Table 3 children-10-00947-t003:** Comparison of the Cobb angles between patient groups with and without complications.

	Complication (−)		Complication (+)		*p*
	Mean ± SD	Median	Mean ± SD	Median	
Cobb Angle (°)					
Preop	61.1 ± 12.1	58.0	68.0 ± 16.5	65.5	0.358 ^t^
Early Postop	19.3 ± 9.7	18.0	22.8 ± 10.3	24.0	7 ^t^
Final Follow-up	23.0 ± 13.9	20.0	27.9 ± 11.5	28.5	7 ^t^
Change Compared to Preop					
Early Postoperative Change	−41.9 ± 10.6	−42.5	−45.3 ± 21.5	−43.5	0.696 ^t^
Intragroup Change *p*	0.000	^E^	0.001	^E^	
Final Follow-up Change	−38.1 ± 11.8	−39.5	−40.1 ± 23.3	−42.0	0.832 ^t^
Intragroup Change *p*	0.000	^E^	0.002	^E^	

^t^ Independent samples *t*-test/^E^ Paired samples *t*-test.

**Table 4 children-10-00947-t004:** Comparison of T1–T12 length between patient groups with and without complications.

	Complication (−)		Complication (+)		*p*
	Mean ± SD	Median	Mean ± SD	Median	
T1–T12 (cm)					
Preop	18.5 ± 2.5	18.9	16.4 ± 2.6	16.6	0.156 ^m^
Early Postop	21.2 ± 2.5	21.4	18.8 ± 2.2	18.9	0.083 ^m^
Final Follow-up	22.3 ± 2.3	23.1	20.3 ± 3.0	19.1	0.156 ^m^
Change Compared to Preop					
Early Postoperative Change	2.7 ± 2.5	2.1	2.4 ± 1.1	2.4	7 ^m^
Intragroup Change *p*	0.018	^W^	0.012	^W^	
Final Follow-up Change	3.8 ± 2.0	3.2	3.9 ± 1.7	3.4	0.916 ^m^
Intragroup Change *p*	0.012	^W^	0.012	^W^	

^m^ Mann–Whitney U test/^w^ Wilcoxon test.

**Table 5 children-10-00947-t005:** Comparison of the T1–S1 length between patient groups with and without complications.

	Complication (−)		Complication (+)		*p*
	Mean ± SD	Median	Mean ± SD	Median	
T1–S1 (cm)					
Preop	30.5 ± 4.0	31.1	26.7 ± 4.5	27.0	0.098 ^t^
Early Postop	34.5 ± 3.2	34.4	30.6 ± 3.6	29.7	0.038 ^t^
Final Follow-up	35.5 ± 3.0	36.0	31.6 ± 3.7	30.5	0.040 ^t^
Change Compared to Preop					
Early postoperative Change	4.1 ± 1.2	3.9	3.9 ± 2.0	3.3	0.882 ^t^
Intragroup Change *p*	0.000	^E^	0.001	^E^	
Final Follow-up Change	5.0 ± 1.5	4.4	5.0 ± 2.3	4.8	0.949 ^t^
Intragroup Change *p*	0.000	^E^	0.000	^E^	

^t^ Independent samples *t*-test/^E^ Paired samples *t*-test.

**Table 6 children-10-00947-t006:** Comparison of the quality-of-life scores measured preoperatively and at final follow-up.

EOSQ-24	Preoperative (Mean ± SD)	Postoperative (Mean ± SD)	*p* Value *
General health	5.13 ± 1.84	7.93 ± 1.33	0.001
Pain	7.06 ± 2.21	8.53 ± 1.180	7
Pulmonary function	6.40 ± 2.09	8.93 ± 1.64	0.001
Mobility	3.13 ± 1.62	4.06 ± 1.09	0.014
Physical function	9.86 ± 4.29	12.20 ± 3.29	0.05
Daily living	6.86 ± 2.94	7.85 ± 2.44	0.539
Energy	6.40 ± 2.92	8.20 ± 1.89	0.011
Emotion	5.60 ± 2.66	7.80 ± 1.78	0.001
Paternal burden	14.00 ± 4.37	18.86 ± 4.67	0.001
Financial burden	2.60 ± 1.21	3.53 ± 1.24	0.010
Satisfaction	7.26 ± 1.83	8.73 ± 1.53	0.004

EOSQ-24: Early Onset Scoliosis Questionnaires-24, SD: Standard deviation. * Paired samples *t*-test.

## Data Availability

The data presented in this study are openly available in [preprints.org] at [https://doi.org/10.20944/preprints202304.0992.v1].

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
