# Peer review of "Implant-Related Complications Do Not Interfere with Corrections with the Shilla Technique in Early Onset Scoliosis: Preliminary Results"

_children, 2023, doi:10.3390/children10060947_

Round 1

Reviewer 1 Report

Dear Authors,

your paper seems really interesting and well structured.

Research about spine surgery for early onset scoliosis is very important, so your efforts are commendable.

Some minor concerns have to be addressed.

In order to improve the visibility of your paper, I suggest to use some keywords different from those mentioned in the title.

Please, indicate the study design in the abstract and clearly in the methods section.

I think that your article should just be integrated in the discussion section, with a brief initial consideration about scoliosis epidemiology.

To do that, I suggest the following references:

-Notarnicola A, Farì G, Maccagnano G, Riondino A, Covelli I, Bianchi FP, Tafuri S, Piazzolla A, Moretti B. Teenagers’ perceptions of their scoliotic curves. an observational study of comparison between sports people and non- sports people. Muscles Ligaments Tendons J [Internet]. 2019;9(2):225-35.

-AlNouri M, Wada K, Kumagai G, Asari T, Nitobe Y, Morishima T, Uesato R, Aoki M, Ishibashi Y. The incidence and prevalence of early-onset scoliosis: a regional multicenter epidemiological study. Spine J. 2022 Sep;22(9):1540-1550. doi: 10.1016/j.spinee.2022.03.016. Epub 2022 Apr 3. PMID: 35381360.

Best regards and good luck

Author Response

Report 1

Dear Authors,

your paper seems really interesting and well structured.

Research about spine surgery for early onset scoliosis is very important, so your efforts are commendable.

Some minor concerns have to be addressed.

In order to improve the visibility of your paper, I suggest to use some keywords different from those mentioned in the title.

Answer: Changes were made accordingly and highlighted.

Please, indicate the study design in the abstract and clearly in the methods section.

Answer: Necessary additions were made about the design of the study.

I think that your article should just be integrated in the discussion section, with a brief initial consideration about scoliosis epidemiology.

To do that, I suggest the following references:

-Notarnicola A, Farì G, Maccagnano G, Riondino A, Covelli I, Bianchi FP, Tafuri S, Piazzolla A, Moretti B. Teenagers’ perceptions of their scoliotic curves. an observational study of comparison between sports people and non- sports people. Muscles Ligaments Tendons J [Internet]. 2019;9(2):225-35.

-AlNouri M, Wada K, Kumagai G, Asari T, Nitobe Y, Morishima T, Uesato R, Aoki M, Ishibashi Y. The incidence and prevalence of early-onset scoliosis: a regional multicenter epidemiological study. Spine J. 2022 Sep;22(9):1540-1550. doi: 10.1016/j.spinee.2022.03.016. Epub 2022 Apr 3. PMID: 35381360.

Answer: After thorough review, the literature recommendations were incorporated to the study's discussion section. (Reference: Notarnicola et al., 2019; AlNouri et al., 2022).

Best regards and good luck

Answer: The researchers thank you for your valuable comments and contributions.

Reviewer 2 Report

This article presents a retrospective analysis of shilla for EOS. 

Main issues:

-short follow up, to be added as a limit

-background, analysis and discussion about the type of proximal fixation and causes of failure should be improved.

Specific comments:

-revise English;

-add "short follow-up" or "early" or "preliminary" to the title

-add some lines about proximal fixation in background and discussion

-precise if the cohort is from one center and briefly describe it (tertiary,  academic, pediatric hospital...)

-I do not understand the sentence "Although 3 patients (18.8%) 79 were graduated". 

-Is kyphosis measured between T4 and T12? lordosis L1-S1? please precise

-Line 88, Sanders stage is repeated 

-Could you suggest some factors related to proximal failure? maybe add some analyses?  

English is not perfect

Author Response

This article presents a retrospective analysis of shilla for EOS. 

Main issues:

-short follow up, to be added as a limit

Answer: Additions have been made to the limitation section.

-background, analysis and discussion about the type of proximal fixation and causes of failure should be improved.

Answer:  By accessing the necessary information and the literature that explained the proximal failure, explanations were added to the discussion section.

Specific comments:

-revise English;

Answer: The English editing service reorganized our content in accordance with your suggestions to give it a better English structure (https://www.mdpi.com/authors/english). (These parts were not highlighted) 

-add "short follow-up" or "early" or "preliminary" to the title

Answer: Changes were made accordingly and highlighted.

-add some lines about proximal fixation in background and discussion

Answer:  In the discussion section, issues with proximal fixation were raised.

-precise if the cohort is from one center and briefly describe it (tertiary,  academic, pediatric hospital...)

Answer: The investigation was conducted in three distinct academic hospitals under the direction of the responsible surgeon. This has been added to the relevant results section.

-I do not understand the sentence "Although 3 patients (18.8%) 79 were graduated". 

Answer: The ambiguity in this sentence has been cleared. Graduation was specified as fusion surgery, which is the final treatment stage for EOS patients.

-Is kyphosis measured between T4 and T12? lordosis L1-S1? please precise

Answer: Additions have been made. We had made the measurements as you specified.

-Line 88, Sanders stage is repeated 

Answer: Corrected

-Could you suggest some factors related to proximal failure? maybe add some analyses?  

Answer:

-References pertaining to the issues with proximal fixation that we discussed in the discussion section were listed and discussed. (Reference: The effect of distraction-based growth-friendly spinal instrumentation on growth in early-onset scoliosis. Acta Orthop Belg. 2016, 82, 715-722. PMID: 29182111.) (Reference: El-Hawary R.; Sturm P.; Cahill P.; Samdani A.; Vitale M.; Gabos P.; Bodin N.; d'Amato C.; Harris C.; Al Khudairy A.; Smith JT. What is the Risk of Developing Proximal Junctional Kyphosis During Growth Friendly Treatments for Early-onset Scoli-osis? J Pediatr Orthop. 2017, 37, 86-91. DOI: 10.1097/BPO.0000000000000599. PMID: 26192880.)

-I appreciate your caution and guidance very much. I would be very grateful if you could advise any sections I should add or change.

Sincere regards.

Reviewer 3 Report

This is an interesting paper on the results and complications of a (unfortunately small!!) case series of EOS patients treated with the Shila technique.

The series is thoroughly described, especially postoperative complications. The complication rate was remarkably high despite the short follow-up time (median 17 months).

Undoubtedly, the information provided by the paper will be of interest to experts dealing with this complex group of patients.

Author Response

Thank you so much for your helpful suggestions and your crucial conclusions on our paper. I'd like to say that it will serve as a guide for my upcoming writings. Service was obtained from https://www.mdpi.com/authors/english for a higher level of English. I appreciate your valuable comments.

Round 2

Reviewer 2 Report

Thank you for revising. 

I suggest some minor corrections as follows:

-I suggest writing ":" instead of "," in the title: Implant-Related Complications Do Not Interfere with Corrections with the Shilla Technique in early onset Scoliosis: Preliminary results.

-Abstract: Please add data about comparison of correction between patients with complications and without complications. You wrote this in the conclusion but not in results. 

-Methods and results: the sentence "Intra- and inter-correlations were high (intraclass correlation coefficient >0.80, 128 p<0.05)." should go to results; section the sentence "The investigation was conducted in three distinct academic hospitals under the di- 132 rection of the responsible surgeon." should go to methods section.

-Resuts: I suggest writing the exact p-values instead of p<0.05 or p>0.05.

-Discussion: in your series, there are many PJK or proximal fixation failures. Should you suggest that other types of fixation should decrease this phenomenon? For instance, Miladi et al. found less mechanical complications with another system of proximal fixation in a growth-friendly scoliosis surgery:  Minimally Invasive Surgery for Neuromuscular Scoliosis: Results and Complications in a Series of One Hundred Patients. Miladi L, Gaume M, Khouri N, Johnson M, Topouchian V, Glorion C. Spine (Phila Pa 1976). 2018 Aug;43(16):E968-E975. doi: 10.1097/BRS.0000000000002588.

Author Response

I suggest some minor corrections as follows:

-I suggest writing ":" instead of "," in the title: Implant-Related Complications Do Not Interfere with Corrections with the Shilla Technique in early onset Scoliosis: Preliminary results.

Answer: Title corrected as you stated. "Implant-Related Complications Do Not Interfere with Corrections with the Shilla Technique in early onset Scoliosis: Preliminary results."

-Abstract: Please add data about comparison of correction between patients with complications and without complications. You wrote this in the conclusion but not in results. 

Answer: The abstract section now includes the relevant sentence. "In terms of spinal growth and deformity correction, there were no significant differences between patients with implant-related problems and individuals without" occurrences. 

-Methods and results: the sentence "Intra- and inter-correlations were high (intraclass correlation coefficient >0.80, 128 p<0.05)." should go to results; section the sentence "The investigation was conducted in three distinct academic hospitals under the di- 132 rection of the responsible surgeon." should go to methods section.

Answer: Intra- and inter-correlations were high (intraclass correlation coefficient >0.80, p<0.05)." The sentence was added to the end of the first paragraph in the results section.

"The investigation was conducted in three distinct academic hospitals under the direction of the responsible surgeon." sentence placed after the first sentence in the Materials and methods section (line 83,84).

-Resuts: I suggest writing the exact p-values instead of p<0.05 or p>0.05.

Answer: The exact p-values were written as you specified.

-Discussion: in your series, there are many PJK or proximal fixation failures. Should you suggest that other types of fixation should decrease this phenomenon? For instance, Miladi et al. found less mechanical complications with another system of proximal fixation in a growth-friendly scoliosis surgery:  Minimally Invasive Surgery for Neuromuscular Scoliosis: Results and Complications in a Series of One Hundred Patients. Miladi L, Gaume M, Khouri N, Johnson M, Topouchian V, Glorion C. Spine (Phila Pa 1976). 2018 Aug;43(16):E968-E975. doi: 10.1097/BRS.0000000000002588.

Answer:  There is limited information on the success rates of different treatment options for proximal junctional problems in EOS. Several factors can affect the success rates of treatment for proximal junctional problems in EOS. The location of the upper instrumented vertebra in relation to the sagittal apex is an important factor that can affect the risk of PJK [29]. We also experienced a high apex kyphosis causing upper anchor problems in one particular case (Deletion syndrome). Additionally, the technique used for instrumentation can also affect the success rates of treatment [30]. Miladi et al. found fewer mechanical complications with fusionless double ends technique using hooks instead of transpedicular screws for proximal fixation in a growth-friendly scoliosis surgery [29].  Other factors that can affect the success rates of treatment include the severity of the condition, the age of the patient, and the presence of any underlying conditions especially in neuromuscular origin [28, 31] Further research is needed to determine the success rates of different treatment options for proximal junctional problems in EOS.

  1. Miladi L.;, Gaume M.; Khouri N.; Johnson M.; Topouchian V.; Glorion C. Minimally Invasive Surgery for Neuromuscular Scoliosis: Results and Complications in a Series of One Hundred Patients. Spine (Phila Pa 1976). 2018, 43, E968-E975. DOI: 10.1097/BRS.0000000000002588.
  2. Anari J.; Tuason D.; Flynn J.; Akbarnia, B. Instrumentation Strategies for Early Onset Scoliosis: Current Concept Review. Journal of the Pediatric Orthopaedic Society of North America. 2021, 3, 3. DOI:https://doi.org/10.55275/JPOSNA-2021-316
  3. Yang B.; Xu L.; Zhou Q.; Qian Z.; Wang B.; Zhu Z.; Qiu Y.; Sun X. Relook into the Risk Factors of Proximal Junctional Kyphosis in Early Onset Scoliosis Patients: Does the Location of Upper Instrumented Vertebra in Relation to the Sagittal Apex Matter? Orthop Surg. 2022, 14, 1695-1702. DOI: 10.1111/os.13380.

I am updating the article with the final edits you request and making the necessary changes. We appreciate all of your comments and recommendations. I sincerely hope that our preparations were carried out in accordance with your recommendations.
